# B.Y.O. Bees: Managing wild bee biodiversity in urban greenspaces

Maggie Anderson[1,2], Floréal Crubaugh[1], Cady Greenslit[1], Emily Hill[1], Heidi Kroth[1], Emily Stanislawski[1], Relena Ribbons[1,3], Israel Del Toro[1] *

1 Department of Biology, Lawrence University, Appleton, WI, United States of America, 2 Department of Ecology, Evolution, and Behavior, University of Minnesota, Saint Paul, MN, United States of America, 3 Department of Geosciences, Lawrence University, Appleton, WI, United States of America

* Israel.deltoro@lawrence.edu

**Data Availability Statement:** All relevant data are within the paper and its Supporting Information files.

**Funding:** The authors received no specific funding for this work.

## Abstract

As cities become more populated and the density of urban development increases, local biodiversity is threatened. Urban greenspaces have the capacity to preserve pollinator biodiversity, but the quality of support they provide depends on greenspace landscape attributes, including the availability of pollinator habitat and foraging resources. Wild native bees provide important pollination services to urban ecosystems, yet relatively little is known about how urban landscape management influences pollinator community composition and diversity. Our study explores how wild bee communities are affected by greenspace and landscape-level features like pollinator management practices, in urban greenspaces in and around Appleton Wisconsin: a mid-sized urban community spanning more than 100 sq. km. We sampled and identified native bees periodically between late-May 2017 and mid-September of 2018 using standardized arrays of pan traps at 15 sites around the city. We classified greenspaces based on their level of development (urban or suburban) and whether they were managed or unmanaged for increasing wild pollinator diversity. We quantified floral species diversity, floral color diversity, tree species diversity, and proximity of sites to open water for each site and used remotely sensed satellite data from both the USGS National Land Cover Database (NLCD) and the Normalized Difference Vegetation Index (NDVI). All variables were tested as potential correlates of wild bee abundance and species richness. Active pollinator management sites supported higher levels of bee abundance and richness. Notably, active greenspace management (e.g. planting native wildflowers) was a stronger correlate of bee abundance and richness than greenspace size and other landscape-level attributes. Within-greenspace attributes such as floral diversity, tree diversity, and proximity to open water contributed positively to both bee abundance and richness. Based on these findings, we suggest that urban greenspaces may be managed more efficiently and cost-effectively by focusing resources on active management by planting wildflowers, removing invasive species, creating nesting habitat, and providing water resources, rather than simply expanding in area.

**Competing interests:** The authors have declared that no competing interests exist.

## Introduction

As cities become more populated and urban development increases, promoting biodiversity conservation in urban systems becomes increasingly important. Recent decades have seen worldwide declines in wild bee diversity corresponding with increasing human population and the growth of urban areas [1]. However, raised awareness of wild bee biodiversity among urban communities can support initiatives for conservation and urban greenspace management. Often, urbanization leads to the fragmentation of natural areas, which reduces wild bee species richness and drastically alters bee community composition [2]. Conserving urban populations of wild bees is essential not only for the sake of conserving biodiversity and protecting threatened species but because pollination is crucial for food production in urban gardens and nearby agricultural systems [3, 4]. Urban environments have been recognized as a type of habitat with the potential to support healthy and diverse bee populations if properly managed [5].

Several recent studies have explored the effects of enhancing urban habitats for managing pollinator biodiversity [6–8]. In general, bee diversity increases relative to the amount of natural habitat available, although the studies which support this claim in urban areas are highly varied. For example, species in the genus *Bombus* benefit greatly from local gardens and urban green areas, although the quality of habitat in the surrounding urban matrix also plays a key role in their abundance [9]. Some bee species, such as small-bodied cavity nesters, are urban specialists and thrive in cities compared to more rural ecosystems [10]. In this case, high numbers of urban specialists can result in few differences in diversity between rural and urban areas. Finally, a study of Chicago green roofs found that native bee richness and abundance increased in response to green roofs, but only if they contain high-quality foraging resources [11]. These studies agree that although urban development often reduces the overall biodiversity of wild bees, managing urban greenspaces for pollinators may counteract these effects and help to restore wild bee communities. However, the approach and spatial scale of management applications needed to bolster urban bee diversity remains largely unexplored.

Management practices in urban ecosystems strongly affect the biodiversity of urban greenspaces. Green infrastructure such as parks, corridors, urban gardens, and urban forests can be designed to support and conserve wild bees more effectively when more is known about the habitat covariates which promote species diversity [10, 12]. Greenspaces can even be designed to attract and conserve specific wild bee species when specific vegetation is planted and nesting habitat is increased [13–15]. Pollinator-focused management of urban areas requires that habitat and foraging resources are both abundant and accessible throughout the season. For example, a study conducted in alpine meadows found that more heterogeneous landscapes with areas of wet meadow supported higher flower diversity (including many late-blooming species), which sustained bumblebee communities for a longer period compared with less-complex landscapes [16]. However, little research to date has specifically examined the effects of pollinator habitat management practices on wild bees in urban and suburban areas across seasons.

Wild bees are surprisingly diverse in North America. In the United States alone, more than 4000 bee species have been documented, with over 500 species present in Wisconsin. There is a growing interest in the study of wild bees and their role in urban pollination [11, 17–19]. Most North American flowering plant species depend on pollinators [1, 20] and those pollinators are consequently considered keystone species in many ecosystems [21, 22]. Specific research is needed to understand how the abundance and richness of wild bee communities can be improved in the context of urban environments.

Here we investigate the effects of bee habitat management practices across both urban and suburban greenspaces with the goal of evaluating which greenspace attributes best predict wild bee richness and abundance. We expected that large areas with active pollinator habitat

management, including planted floral resources and available nesting habitat, have the most diverse and abundant bee communities.

## Materials and methods

### Study region and experimental sites

Our study took place in the Fox Cities area (population ~250,000 as of 2016) of northeastern Wisconsin (44.263 N, -88.407 W) which includes the cities of Appleton, Menasha, Fox Crossing, Kimberly Kaukauna, and Neenah. The urban region is surrounded by a mix of deciduous forest, natural grassland, pasture, and cropland. A total of 16 sites (Fig 1) were chosen in urban and suburban greenspaces (e.g. parks, preserves, and other natural areas within 15km of Appleton city center) ranging in total area from 200 m$^2$ to 8.0 km$^2$. Larger greenspaces (total area > 5 sq. km) contained two sites, in which case the sites were set up at a minimum of 0.75 km apart. The region has a spring-summer mean annual temperature range from 10˚C-22˚C and a mean annual precipitation of 845 mm. We chose sites such that our sampling was spread over the entire urban and suburban Fox Cities region with approximately 1/3 suburban sites and 2/3 urban sites.

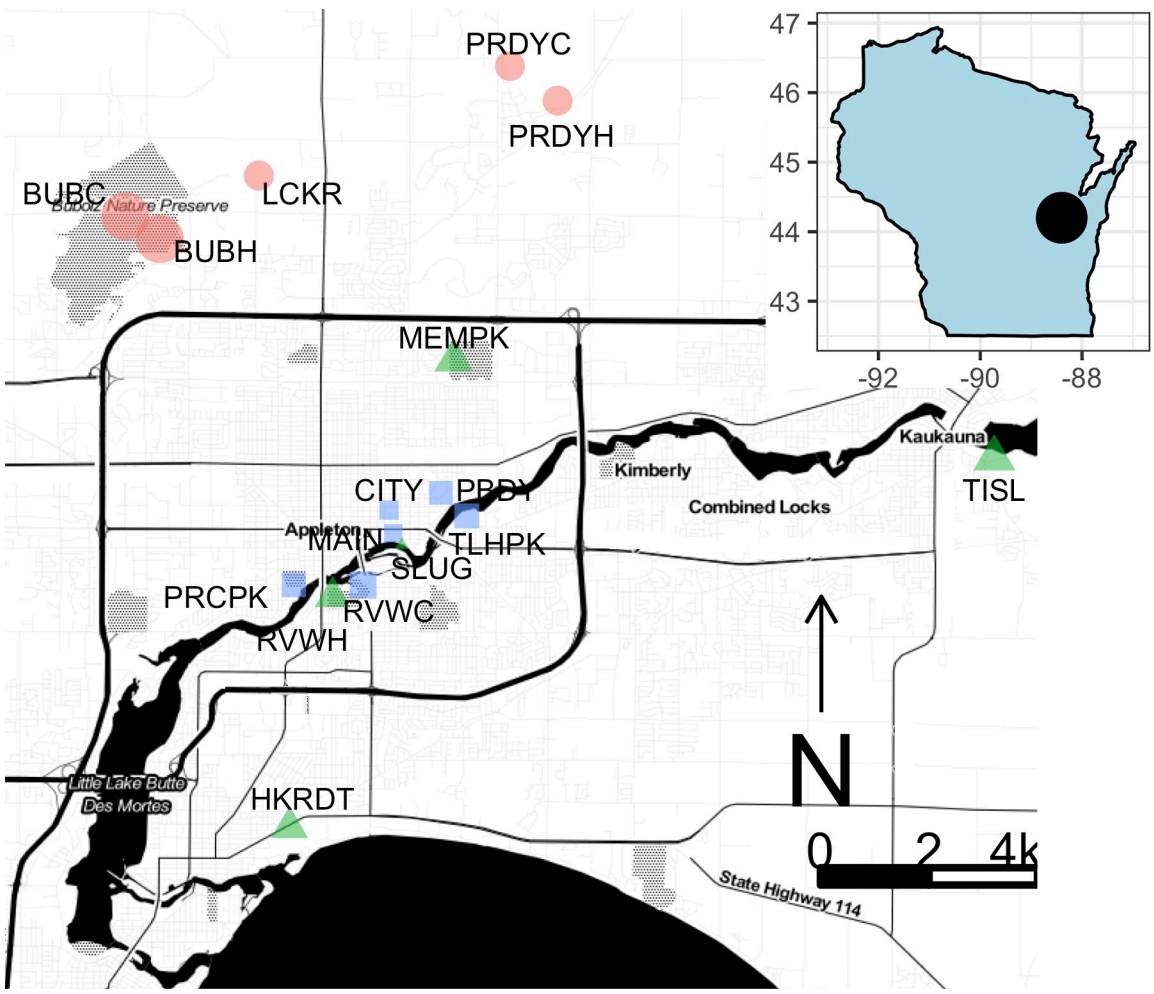

**Fig 1. Map of urban green spaces categorizing of management status and points representing observed bee species richness.** Blue squares are unmanaged urban sites, green triangles are managed urban sites, red points are managed suburban sites. Basemap generated in ggmap [23] using a Stamen basemap (Map tiles by Stamen Design, under CC BY 3.0. Data by OpenStreetMap, under ODbL).

## Management regimes and landscape scale

We define urban areas using the USA Census definition of more than 50,000 residents in a single urbanized cluster, and suburban areas as those with more than 2,500 but fewer than 50,000 residents in an urbanized cluster. We define managed sites as areas that were actively managed for pollinator resource provisioning (i.e., planted with pollinator-friendly seed mixes and/or were improved to create pollinator nesting habitat) while unmanaged sites were not managed with that goal. This method of classification means that although sites were often managed for other purposes, including agriculture, recreation, or aesthetic value, we do not consider them to be "managed" in this study. The 16 sites were classified into three categories: (1) *Unmanaged urban sites*–areas dominated by lawn, constantly mowed with little or no native floral resources and no specific plan for conservation of native biodiversity (n = 6), These sites all had <10% of their total area planted with native flowering plants and were generally dominated by grassy turf (2) *Managed urban sites*–urban areas where native flora or prairies have been established with a conscious effort to restore native biodiversity (these sites also include an increase in nesting habitat availability) (n = 5). These sites all had >25% of their total area planted with native flowering species. (3) *Managed suburban sites*–areas characterized by having lower human population densities and where an active effort is being enacted to restore or maintain native biodiversity (n = 5). These sites also had >25% of their total area planted with native flowering species. We divided all landscape covariates into two scales. We considered landscape-level attributes (greenspace area, USGS National Land Cover Database (NLCD) and the Normalized Difference Vegetation Index (NDVI)) to be large-scale landscape variables with the capacity to apply broadly to the entire greenspace area. Secondly, we considered within-greenspace attributes (management practice, floral diversity, and tree diversity) to be local-scale landscape variables which likely vary at a finer spatial resolution across greenspaces. Because we could not secure sampling permission from county parks, we did not have any unmanaged suburban sites in our study.

## Bee sampling and identification

We used systematic, standardized methods for sampling urban bees. We collected data during the summer (May-September) of 2017 and (June-September) 2018. Specimens were systematically collected at each site using an array of 16 pan trap trays arranged 10m apart in a 4 x 4 grid during every sampling event (see S1 and S2 Figs for images of pan traps and a diagram of the sampling array). Trays were mounted on 1m metal stakes above the surrounding vegetation [24]. Each tray measured 30cm on both sides and held 4 colored plastic pans which were 18cm in diameter and 5cm deep. Pans were colored red (λ ~ 650 nm), yellow (λ ~ 580 nm), blue (λ ~ 475 nm), and white to account for the sensitivity of different species and sexes of bees to pan color [18, 25]. Each pan was treated independently as its own sample. In addition to the pan traps, 5 blue vane traps were used within the array (1 in the center and 4 placed ~14m away from the center in each of the 4 corners) while 4 additional blue vane traps were used around the edge of the grid, each 10m away from the center of each edge. Blue vane traps were used as a means of increasing the sample size as they are effective for trapping different wild bee species in grasslands and prairies [18]. During each sampling event a total of 25 traps (blue vane + pan trap arrays) were deployed. Upon array setup, the pans were filled with 1-2cm of water and dish soap to capture any visiting flying insects.

Each site was sampled with the pan trap array once every two weeks. Although we were unable to sample all 16 sites at once, all sites were sampled within approximately 8–10 days of each other before the sampling round was repeated. We did not sample during inclement weather. At the end of each 12-hour sampling period, specimens were collected in the field,

preserved in ethanol, then taken back to the lab for pinning and identification [10]. Only species belonging to the order Hymenoptera (Apidae) were considered. A representative collection of the specimens is deposited in the Entomological Museum of the University of Wisconsin, Madison. Species IDs were verified by Dr. Jason Gibbs (University of Manitoba Entomology Curator, pers comm.). We used the R package iNEXT [26] to evaluate sample coverage and estimated rarefied species richness. In subsequent analyses we use observed species richness as our response variable rather than estimated species richness, as these two responses are strongly correlated ($R^2 = 0.60$, $P < 0.001$) but observed species is more reflective of known richness at a given site.

## Analyses

Because of the layout of the city of Appleton, there is a clustered distribution of sites. Rural managed sites tended to be in the more undeveloped northern extent of the city and city parks tended to follow the waterways and lakes in the city. Sites were also clustered as a result of discussions with the city landscaping managers, which allowed sampling only at certain urban city parks. In sites not owned by the city of Appleton, we obtained permission from the landscaping managers at each site (e.g. Lawrence University, and Riverview Gardens). We accounted for this potential spatial autocorrelation in our statistical analyses to disentangle the effects of spatial autocorrelation from environmental correlates using the R package lctools [27]. Given the relatively small spatial scale of this study, all bee species have equal potential to colonize any given site given enough time to disperse, as the sites are no more than 16 km apart in distance.

We used a Normalized Difference Vegetation Index (NDVI) [28] to characterize vegetation surrounding each of the study sites [29]. Briefly, NDVI is a graphical indicator used on remote sensing data to assess the abundance and density of green vegetation. We extracted data from a 250m radius of each site using the *MODIS* and *vegan* package in R [30, 31]. We evaluated a radius of 250m, as this was the highest-resolution option for data extraction and because similar analyses have been conducted at comparable distances [9, 23]. We used the mean annual NDVI value for each site from the growing season one year before the sampling event (i.e. April-October 2016–2017). Data from the 2011 National Landcover Database (NLCD) was used to characterize landcover surrounding each of the study sites (https://www.mrlc.gov). Percentages of all landcover types were also extracted within a 250m radius of each site to remain consistent with NDVI data. All data was extracted using R [32] with a total of 13 landcover types identified using Principal Coordinates Analyses (PCoA). In addition to NLCD data, distances of the sites to open water, and roadways were estimated from aerial images to assess the proximity of anthropological disturbance at the sites.

Tree species diversity was assessed at each site within a 20m x 20m plot contained within the 30m x 30m pan trap array plot [33, 34]. Numbers and species of trees in each plot were recorded and relative tree diversity was calculated using the Shannon-Weiner Diversity Index [35]. Floral diversity at each site was evaluated using a randomized sampling of 1m quadrats along two 30-meter perpendicular transect lines. Floral species and color surveys were conducted twice from mid-July to mid-August at each of the sites in the summer of 2018. In each square meter, the number of flowering plants in bloom were counted and species were either identified or photographed and pressed for later identification using dichotomous keys in Wildflowers of Wisconsin and the Great Lakes Region: A Comprehensive Field Guide [36]. We counted the number of blooms and their visible color assessed according to a color wheel [37]. Tree species richness and Shannon-Weiner diversity were calculated for each site.

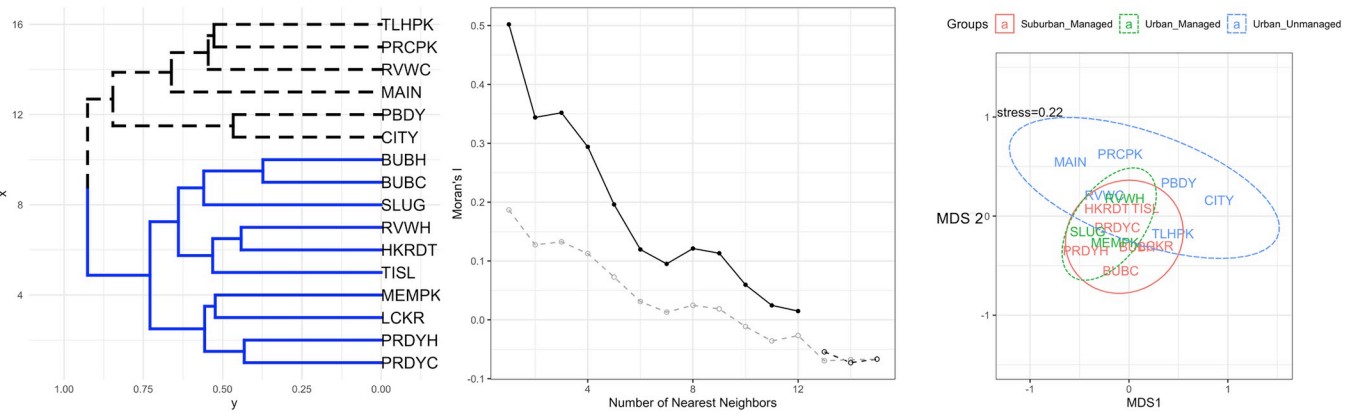

**Fig 2. Bee abundance and species richness across different management practices.** A) Bee abundance across managed suburban, managed urban, and unmanaged urban treatments. B) Bee species richness across managed suburban, managed urban, and unmanaged urban treatments.

## Results

### Wild bee biodiversity

In general, wild bee species richness and community composition increased significantly with management in both urban and suburban areas (Fig 2). In our survey of the urban and suburban greenspaces we identified a total of 1516 individuals belonging to 86 distinct species and 20 genera (S2 Table). We found that sampling coverage for all sites ranged between 0.8 and 0.97 suggesting that we accounted for the majority of species present at the sites. Between 2017 and 2018, each site had 8 sampling dates, with 25 traps at each site. There was a total of n = 200 independent observations per site. At urban unmanaged sites there were a total of 1008 samples collected. In urban managed sites a total of 840 samples were collected. At the suburban managed sites 780 samples were collected. Some traps were lost due to wind, animal and human disturbances, these were omitted from the analyses.

At the landscape level, bee abundance was largely explained by greenspace size and satellite landcover data (Table 1). Greenspaces with high flower species and flower color diversity in addition to high tree diversity were also more likely to support high wild bee abundances (Table 1). Wild bee species richness was primarily correlated by within-greenspace attributes including tree and floral diversity as well as the availability of water resources (Table 1). Using the PCoA, we summarized the 13 landscape variables as the two principal coordinate axes that explained < 95% of the variation in the dataset and were subsequently used as predictor variables in the regression models. The PCoA1 axis differentiated the suburban managed sites primarily with the landscape attributes of percent woody wetlands and cultivated crops. While the PCoA2 axis separated the urban unmanaged as being dominated by medium to high intensity developed sites, but urban managed sites were mainly classified as developed open space. We found that spatial autocorrelation was highest between the three nearest neighbor sites and gradually decreased with increased distance between sites, a pattern that was expected due to the greenspace distribution in the city (Fig 3B).

### Response of bee communities to landscape-level greenspace attributes

Wild bee abundance was significantly predicted by greenspace management and landscape-level attributes. Greenspace area was not a strong predictor of bee abundance alone ($P > 0.05$) but contributed to higher bee abundances when considered along with management practice. Similarly, bee abundances were highly correlated with landscape-level remotely sensed data,

**Table 1. Summary of global and best fit GLMs.** The response variables in each model are either bee species richness or abundances. The global model includes all potential explanatory variables while the best fit model is reduced based on AIC values to only the variables that contribute significantly. The estimate values represent the directionality of the association between the response and predictor variables.

| Model | Response | Variables | Coefficient Estimate | AIC |
|---|---|---|---|---|
| Global Model | | | | 97.98 |
| | Bee Species Richness | | | |
| | | NDVI | 0.045 | |
| | | Management treatment | | |
| | | Managed Urban | -0.0074 | |
| | | Unmanaged Urban | -0.079 | |
| | | Tree species diversity | -0.0081 | |
| | | Distance to water | -1.432 e-04 | |
| | | Green space area | -3.736e-08 | |
| | | NLCD PCo2 | 0.0046 | |
| | | Flower species diversity | -0.017 | |
| Best-fit Model | | | | 90.04 |
| | Species Richness | Distance to water | -0.001 | |
| | | Management treatment | | |
| | | Managed Urban | -0.02 | |
| | | Unmanaged Urban | -0.63 | |
| Global Model | | | | 134.95 |
| | Bee Abundance | | | |
| | | NDVI | -0.048 | |
| | | Management treatment | | |
| | | Managed Urban | -0.050 | |
| | | Unmanaged Urban | -0.19 | |
| | | Tree species diversity | -0.02 | |
| | | Distance to water | -1.055e-03 | |
| | | Green space area | -1.631e-07 | |
| | | NLCD PCo2 | 0.02 | |
| | | Flower species diversity | -0.014 | |
| Best-fit Model | Bee Abundance | | | 131.79 |
| | | Management treatment | | |
| | | Managed Urban | -0.046 | |
| | | Unmanaged Urban | 0.19 | |
| | | Tree species diversity | -0.024 | |
| | | Distance to water | -1.082e-03 | |
| | | Green space area | -1.639e-07 | |
| | | Flower species diversity | -0.014 | |

although only when combined with management practice. Wild bee abundance was also significantly predicted by both landcover and presence of green vegetation in managed greenspaces: NLCD and NDVI.

## Associations between bee communities and small-scale greenspace attributes

Both wild bee abundance and species richness were correlated with greenspace attributes in addition to management practices. Floral diversity was a major predictor of both wild bee abundance and richness. When paired with management style, floral diversity became a stronger predictor of both abundance (AIC = 202.33) and richness (AIC = 99.11). Floral color

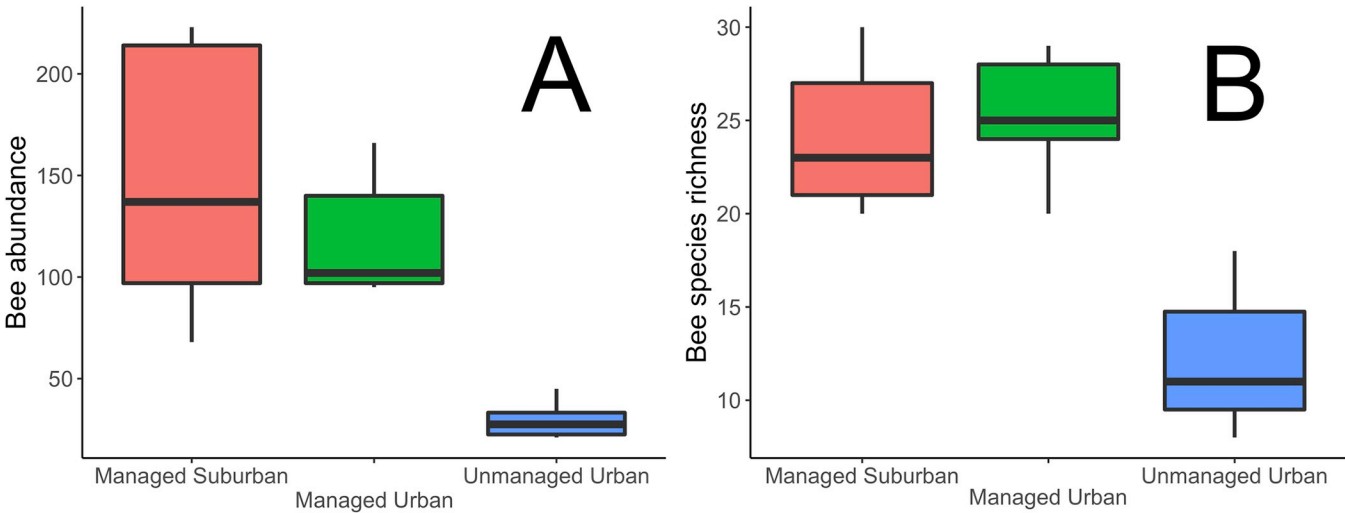

**Fig 3. Beta diversity and community structure of study sites.** A) Bray-Curtis similarity between sites with two natural clusters identified. B) Test of spatial autocorrelation for richness (grey) and abundance (black). Solid lines and points indicate significant spatial autocorrelation effects. Dashed lines and hollow points indicate non-significant spatial autocorrelation effects. C) NMDS ordination plot of the study sties grouped by the three management types with lines indicating 95% confidence intervals. Solid line indicates suburban managed sites, dotted lines indicate urban managed sites and dashed lines indicate urban unmanaged sites.

diversity was also a significant predictor of bee abundances when paired with management practices. Tree diversity also contributed significantly to wild bee abundance and richness, although only when paired with management practices. Finally, proximity of sites to open water was a significant predictor of wild bee species richness but not bee abundance.

We generated a correlogram using the R package corrgram [38] to assess the amount of multicollinearity between these environmental variables and subsequently removed those which were highly correlated ($R^2 \geq 0.75$). The remaining variables were used to generate a generalized linear model (GLM) and assess the relationship between wild bee abundance or species richness and the other environmental variables (Table 1), assuming a Poisson family distribution which is appropriate for count data. The best-fit GLMs for bee abundance and species richness were chosen according to their Akaike's Information Criterion (AIC) values (Table 1) based on stepwise variable selection using the R package mass [39] and the significance of the predictor variables in the best-fitting model was tested using likelihood ratio tests (Table 2).

### Bee community response to management styles

Managed habitats in both urban and suburban areas supported higher wild bee abundance and species richness (Fig 3). Managed sites in both urban and suburban settings supported

**Table 2. Summary of likelihood ratio test (LRT) results for each variable included in the best-fit models for bee species richness bee abundance.**

| Model | Variables | DF | Deviance | AIC | Chi2 | Pr(>Chi) |
|---|---|---|---|---|---|---|
| Best-fit Model (Bee Species Richness) | Treatment | 2 | 28.5715 | 108.836 | 22.7942 | < 0.0001 *** |
| | Distance to water | 1 | 11.2378 | 93.502 | 5.4606 | 0.0195 * |
| Best-fit Model (Bee Abundance) | Treatment | 2 | 308.542 | 416.37 | 288.577 | < 0.0001 *** |
| | Tree species diversity | 1 | 34.693 | 144.52 | 14.727 | 0.0001 *** |
| | Distance to water | 1 | 32.268 | 142.10 | 12.303 | 0.0004 *** |
| | Green space area | 1 | 128.600 | 238.43 | 108.635 | < 0.0001 *** |
| | Flower species diversity | 1 | 23.017 | 132.85 | 3.051 | 0.0807. |

similar wild bee communities compared to unmanaged urban sites. Urban and suburban managed sites did not vary greatly in bee abundance or species richness compared to each other, but bee abundance increased significantly in managed sites compared to unmanaged sites. Please note there were no unmanaged suburban sites for comparison, so unmanaged sites only refer to unmanaged urban areas. Bee species richness was significantly higher in managed compared to unmanaged sites. Wild bee community composition also varied as a function of management practices, with more similar communities observed at managed sites compared to unmanaged sites (Fig 2).

## Discussion

### Correlates of wild bee diversity and abundance

Wild bee abundance and richness are positively affected by active management styles in both urban and suburban areas. At a greenspace-scale, environmental variables were the most effective predictors of higher wild bee abundance and richness. Conservation management in urban and suburban areas may be enhanced by focusing on active management for bees rather than land preservation alone. Our results have the potential to improve bee conservation efforts through the development of greenspace management practices that are more efficient and cost-effective for improving wild bee habitat in urban areas. Doing things like replacing ornamental with native flora, reducing chemical use and mowing frequency all have the potential to attract native wild bees. Active pollinator management can enhance wild bee diversity, in both urban and suburban landscapes, and should be further implemented as best conversation management practice.

Landscape-scale variables were instrumental predictors of bee abundance although not of bee richness. Larger greenspaces likely supported higher abundances of bees because they have more available land area for nesting and foraging. Our results are consistent with previous studies which find that urban bee communities benefit greatly from larger areas of contiguous urban greenspace in the surrounding landscape [11, 40] and that green landscapes with low impervious surface area support greater diversity compared with heavily-urbanized landscapes with few greenspaces [41]. From a conservation standpoint, establishing large urban greenspaces will prove essential in maintaining healthy, diverse bee communities in the long-term [42]. Our finding that actively managed landscapes support greater diversity should be further developed as a modeling and greenspace planning tool. If landscape-level variables are predictive of bee diversity, then it may be possible to estimate large-scale bee abundances based on remotely sensed data.

Although some landscape-level variables were positively associated with wild bee abundance, greenspace-level environmental variables significantly increased both bee abundance and species richness. Floral diversity had a significant positive influence on bee abundance, supporting previous work showing that bee communities benefit greatly from a variety of available foraging resources [43, 44]. Diverse floral resources provide constant forage throughout the summer, as varied and overlapping bloom periods ensure that sufficient forage is always available [45]. Floral color diversity was also a key correlate a bee species abundance, although not of richness. Little research to date has investigated the relationship between floral color diversity and wild bee communities, although bee visitation to colored pan traps suggests that multiple colors attract a greater diversity of bees [46]. Similarly, trees provide bees with valuable nesting habitat resources; our research corroborates previous work which finds a positive effect of tree diversity on bee abundance [47]. Many species of wild bees are cavity-nesters or rely otherwise on trees for floral resources [10] and it has been suggested that trees–especially flowering trees–be planted in urban greenspaces to preserve local pollinator communities [48].

We also found that wild bee abundance and richness was strongly associated with proximity to open water. To our knowledge, ours is the first study to find a positive indirect relationship between urban bee communities and water resources. This relationship may be explained by the tendency of lakes, rivers, and streams to be associated with wet meadows and similar low-lying areas suitable for a variety of herbaceous plant species [49] which in turn support more diverse pollinator communities [50]. Wet meadows also provide more floral resources in late summer, making them a critical habitat for many overwintering bee species [16]. Although exploring this relationship further is beyond the scope of our study, we recommend that future studies specifically investigate the effects of wildflower diversity on pollinator communities in urban wet areas. From a conservation standpoint, it is likely that adding a source of open water or wet meadow habitats to greenspaces would diversify the palate of wildflowers available to wild bees and increase the capacity of greenspaces–especially small ones–to support more diverse wild bee communities.

### Recommendations for greenspace management

We identify active management of greenspaces as the most consistent positive influence on wild bee abundance and diversity in urban areas. Unmanaged urban greenspaces exhibited the lowest levels of bee abundance and diversity, indicating the traditional management approach does not promote the diversity of urban wild bee communities. It should be noted that we only considered managed suburban, managed urban, and unmanaged urban sites in this study. The potential for unmanaged suburban sites or otherwise passively managed greenspaces to support pollinator communities warrants further study. Work by Twerd and Banaszak-Cibicka [51] found "urban wastelands" provided important secondary habitats for bees, where grassy suburban unmanaged sites and reclaimed sand and clay pits were especially attractive habitats for bees.

Greenspace-scale variables like floral diversity and water resources were the most important predictors of both bee abundance and richness. This indicates that greenspace managers should prioritize the active improvement of existing greenspaces for wild bees rather than simply preserving large areas. For example, one of our smallest and most urban sites (SLUG) exhibited one of the highest levels of wild bee diversity, especially compared to other urban sites. Notably, this site is an urban garden which has been planted with a variety of wildflowers, fruit trees, and native flower mixes and is adjacent to a water source. Although large greenspaces promote high bee abundance, they must be managed further to support high bee diversity. Greenspaces can be managed to benefit wild bee communities by planting floral resources, planting trees, and adding sources of open water or wetland habitats.

Planting native prairie mixes within greenspaces is one cost-effective way to increase foraging resources for wild bees, one that can be implemented easily in most urban areas of the Midwest USA. Numerous studies demonstrate that native prairie plantations greatly benefit bee communities [52–54]. Prairie mixes are also perennial, making them easy for cities to manage in the long-term. We also recommend adding water resources such as ponds or wet meadow habitat to further improve greenspaces for wild bees. Although such physical changes to greenspaces are often costly to implement, our study supports a growing body of evidence that water resources support high-quality forage for wild bees. However, additional experimental research is needed to firmly establish a direct connection between pollinator communities and urban water resources. Numerous other studies also demonstrate a strong positive association between patch size and both bee abundance and species richness [55, 56]. Our study supports these findings, indicating that a square kilometer of greenspace will support approximately 20 different species of bees (see supplementary data and code). From a conservation standpoint, this number could be incorporated into future urban planning projects.

## Conclusions

We demonstrate that active management of urban greenspaces supports a greater diversity and abundance of wild bees than unmanaged urban spaces. Actively managed greenspaces such as urban gardens, native prairie plantations, and wet habitats were associated with more diverse bee communities with observable differences between urban and suburban areas. We note that our results do not reflect potential diversity patterns in unmanaged suburban sites. Future research could address the relationship between pollinator diversity and traditionally managed landscapes that are not focused on pollinators; thus, adding a missing piece of knowledge about these unmanaged suburban sites into pollinator biodiversity assessments. We recommend that future urban management focus on actively improving greenspaces for wild bees (i.e. by planting flowers) regardless of greenspace size or position within the urban matrix. Although habitat fragmentation in urban areas negatively affects bee diversity, we demonstrate that many of these effects can be remedied through the active management of urban greenspaces for wild bees. This is consistent with a meta-analysis by Winfree et al. [20] who found that habitat fragmentation significantly negatively influenced bee populations, when in systems with little natural habitat remaining. Urban spaces managed for pollinator conservation can alleviate this negative influence, by providing a variety of habitat and floral resources for bees, which is supported by this study and other recent work on urban bee populations [57, 58].

Further research on this topic should consider additional habitat vectors, especially wildflower abundance and presence at sites throughout the growing season. As we continue to develop this large dataset, we hope to investigate temporal trends and explore the effects on ongoing urban management on urban bee communities. Native prairie cover at sites could be quantified and wet habitats could be further classified, to determine if sites with wet meadows did in fact promote higher bee species richness, as speculated. Native prairie and water source installation could also be used as experimental treatments in a long-term study. Sites could be classified along an urban gradient determined by percent impervious surfaces (see [10]) to investigate the extent to which the intensity of urbanization is negated by access to adequate floral resources. We also highlight the need for data from unmanaged suburban greenspaces as these may be reflective of intermediate richness and turnover patterns between managed urban and suburban habitats. Large-scale studies could also be conducted in downtown metropolitan areas to determine if these conservation applications are still effective in much larger cities. Future research of this type on wild bees offers an exciting opportunity to quantify the value of pollinator ecosystem services in urban landscapes.

## Supporting information

**S1 Table. Site bee abundances.** Summary of site bee abundance, richness and sampling effort. (DOCX)

**S2 Table. Wild bee species abundances.** Summary of total abundances of all wild bee species collected across all sites.
(DOCX)

**S1 Fig. Schematic of field sampling design.** Photos of (A) a multicolored pan trap and (B) a blue vane trap at a field site. Photos: M. Anderson.
(PNG)

**S2 Fig. Pan trap sample array.** Diagram of the pan trap sample array showing the positions of multicolored pan traps and blue vane traps.
(PNG)

**S3 Fig. Total bee abundance from 2017–2018.** Total bee abundance observed at each site over the course of two field seasons from 2017–2018.
(PNG)

**S1 File.**
(R)

## Acknowledgments

We would like to thank Hailey Bomar, Linder Wendt, and the other members of the BYO-BEEZ research team for their help with research and data collection. Additionally, we would like to acknowledge Lawrence University, The City of Appleton, Riverview Gardens, Gordon Buboltz Nature Preserve, Bruce B. Purdy Nature Preserve, Heckrodt Wetland Preserve, and Thousand Island Nature Preserve for their cooperation in granting access to research sites for this project.

## Author Contributions

**Conceptualization:** Relena Ribbons, Israel Del Toro.

**Data curation:** Floréal Crubaugh, Cady Greenslit, Heidi Kroth, Israel Del Toro.

**Formal analysis:** Maggie Anderson, Cady Greenslit, Heidi Kroth, Emily Stanislawski, Relena Ribbons, Israel Del Toro.

**Funding acquisition:** Israel Del Toro.

**Investigation:** Maggie Anderson, Floréal Crubaugh, Cady Greenslit, Emily Hill, Heidi Kroth, Emily Stanislawski, Relena Ribbons, Israel Del Toro.

**Methodology:** Maggie Anderson, Floréal Crubaugh, Cady Greenslit, Emily Hill, Emily Stanislawski, Relena Ribbons, Israel Del Toro.

**Project administration:** Relena Ribbons, Israel Del Toro.

**Resources:** Relena Ribbons.

**Supervision:** Relena Ribbons, Israel Del Toro.

**Validation:** Israel Del Toro.

**Visualization:** Maggie Anderson, Floréal Crubaugh, Cady Greenslit, Emily Hill, Heidi Kroth, Emily Stanislawski, Israel Del Toro.

**Writing – original draft:** Maggie Anderson, Floréal Crubaugh, Cady Greenslit, Emily Stanislawski, Relena Ribbons.

**Writing – review & editing:** Relena Ribbons, Israel Del Toro.

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
