## [Decision Letter · Decision Letter 0]

18 Nov 2021

PONE-D-21-32723B.Y.O. Bees: managing wild bee biodiversity in urban greenspacesPLOS ONE

Dear Dr. Del Toro,

Thank you for submitting your manuscript to PLOS ONE. After careful consideration, we feel that it has merit but does not fully meet PLOS ONE’s publication criteria as it currently stands. Therefore, we invite you to submit a revised version of the manuscript that addresses the points raised during the review process.

This article has been reviewed by two reviewers, which found the subject interesting and relevant for the readers of Plos One. However, they have important concerns regarding the clarity of methodology and the analyses. If you decide to revise the manuscript, please send us a letter with point-by-point responses to each particular comment. The manuscript will be then send out for review again to the same or other reviewers.

We look forward to receiving your revised manuscript.

Kind regards,

Amparo Lázaro, PhD

Academic Editor

PLOS ONE

Journal Requirements:

2. Please note that PLOS ONE has specific guidelines on code sharing for submissions in which author-generated code underpins the findings in the manuscript. In these cases, all author-generated code must be made available without restrictions upon publication of the work. Please review our guidelines at https://journals.plos.org/plosone/s/materials-and-software-sharing#loc-sharing-code and ensure that your code is shared in a way that follows best practice and facilitates reproducibility and reuse. Code may be shared by providing a URL within the Methods section to a code repository or it may be uploaded as a supplemental file.

4. Please amend the manuscript submission data (via Edit Submission) to include authors Maggie Anderson, Floréal Crubaugh, Cady Greenslit, Emily Hill, Heidi Kroth, Emily Stanislawski, Relena R. Ribbons.

5. We note that Figure 1 in your submission contain copyrighted images. All PLOS content is published under the Creative Commons Attribution License (CC BY 4.0), which means that the manuscript, images, and Supporting Information files will be freely available online, and any third party is permitted to access, download, copy, distribute, and use these materials in any way, even commercially, with proper attribution. For more information, see our copyright guidelines: http://journals.plos.org/plosone/s/licenses-and-copyright.

a) You may seek permission from the original copyright holder of Figure 1 to publish the content specifically under the CC BY 4.0 license. 

6. Please include a copy of Table 2 which you refer to in your text on page 8.

Reviewers' comments:

Reviewer's Responses to Questions

**Comments to the Author**

1. Is the manuscript technically sound, and do the data support the conclusions?

Reviewer #1: Yes

Reviewer #2: Yes

2. Has the statistical analysis been performed appropriately and rigorously? 

Reviewer #1: Yes

Reviewer #2: I Don't Know

3. Have the authors made all data underlying the findings in their manuscript fully available?

Reviewer #1: Yes

Reviewer #2: Yes

4. Is the manuscript presented in an intelligible fashion and written in standard English?

Reviewer #1: Yes

Reviewer #2: Yes

5. Review Comments to the Author

Reviewer #1: This paper presents the results of a two-year research evaluating bees in urban and suburban habitats. Urban habitats were both managed and unmanaged ones. Whereas, suburban habitats were exclusively the managed ones. Management was found to increase bee richness and abundance. In my view, this is an overstatement as the unmanaged suburban areas were not subject to an analysis. Thus, this conclusion needs to be made more precise.

Main impressions and journal's standards

Generally, the paper was well written and is very interesting to read. The topic is important for a wide audience and the findings can be quite useful, e.g. for urban planners or natural resource managers. The analysis is mostly sound and the conclusions justified. There are, however, some issues to resolve and I would thus recommend the manuscript for revision before it can be published.

Materials and methods

Management regimes and landscape scale.

Was there a management level, i.e., what % of the area was sown with pollinator-friendly plants? What % of the area was designated to fulfil this criterion? Could you briefly describe what plant species were dominant in the unmanaged habitats? Please, explain why unmanaged suburban areas were not included in the research conducted.

Bee sampling and identification.

Information on the total number of samples taken in each year of the study and in the urban and suburban habitats would be useful.

For the purpose of giving other scientists an idea of the climatic conditions in the research area, please add information on what average temperatures prevail in the area during springtime and summertime period.

Each site was sampled with the pan trap array once every two weeks.....

Define precisely what the trial means to you. The sum of bees caught in all bowls? Was this data merged?

Analyses

Tree species diversity was assessed at each site within a 20m x 20m plot contained within the 30m x 30m pan trap array plot .....

Were the flower colours of woody plants also assessed with the use of colour wheel? Why no assessment of the bee nutrient base in the springtime was conducted? Could you additionally include a list of plants with their assigned flower colours in the appendix?

Results

Bee community response to management styles.

Formulate your conclusions carefully. The sentence above may be misleading as the unmanaged suburban habitats were not included in the research.

Discussion

Although some landscape-level variables were positively associated with wild bee ...

What is the plants base in the springtime like? I am missing this information.

Recommendations for greenspace management.

You should also refer your results to the literature which indicates that the unmanaged habitats are also attractive to wild bees. Unmanaged sites, rich in the floral species also provide stable resources.

Twerd L., Banaszak-Cibicka W. 2019. Wastelands: their attractiveness and importance for preserving the diversity of wild bees in urban areas. Journal of Insect Conservation 23(3): 573–588.

Machon N. 2021. Urban Wastelands Can Be Amazing Reservoirs of Biodiversity for Cities. Urban wastelands 3-18.

McKinney. 2021. Strategies for Increasing Biodiversity Conservation in Cities Using Wastelands: Review and Case Study. Urban wastelands 39-64.

Twerd L., Banaszak-Cibicka W., Sobieraj-Betlińska A., Waldon-Rudzionek B., Hoffmann R. 2021. Contributions of phenological groups of wild bees as an indicator of food availability in urban wastelands. Ecological Indicators 126: 107616.

Reviewer #2: Review for submitted manuscript entitled ‘B.Y.O. Bees: managing wild bee biodiversity in urban greenspaces’.

The manuscript focuses on how wild bee communities are affected by greenspace and landscape-level features in urban greenspaces. How pollinators cope with urbanization is now a very important issue. The subject of the paper is therefore interesting. This is not a new issue, but an additional point of discussion to the current literature. While these results are not especially surprising, they do help policy makers and urban planners by providing compelling evidence for advisability of appropriate urban management for those insects.

My primary concerns with this study have to do with the study design. The paper lacks specific information on methodology. More details about the study sites should be provided. It could be done in the table or in a more descriptive manner. What were the management practices for pollinators, floral diversity, tree diversity on individual sites?

Justification of three study site types selection (managed urban sites, unmanaged urban sites and managed suburban sites) is needed. Why are unmanaged suburban sites missing?

How many samples were collected for each site type?

Add reference to the method floral diversity evaluation at each site.

There are no figures for PCoA analyzes.

Information on bee diversity is missing in the results, but appears in the discussion for the first time in the paper.

“Although habitat fragmentation in urban areas negatively affects bee diversity, we demonstrate that many of these effects can be remedied through the active management greenspaces for wild bees.” - was it analyzed at this paper? I don’t believe that such analyzes are mentioned in the methods section.

Overall, there are several points that need to be changed and improved until this paper is ready for publication.

6. PLOS authors have the option to publish the peer review history of their article (what does this mean?). If published, this will include your full peer review and any attached files.

Reviewer #1: No

Reviewer #2: No

---

## [Author Response · Author response to Decision Letter 0]

3 May 2022

Please see attached Response to Reviewer Document for detailed comments, thank you.

---

## [Decision Letter · Decision Letter 1]

13 Jul 2022

PONE-D-21-32723R1B.Y.O. Bees: managing wild bee biodiversity in urban greenspacesPLOS ONE

Dear Dr. Del Toro,

Thank you for submitting your manuscript to PLOS ONE. After careful consideration, we feel that it has merit but does not fully meet PLOS ONE’s publication criteria as it currently stands. Therefore, we invite you to submit a revised version of the manuscript that addresses the points raised during the review process.

This manuscript have been now reviewed by one of the previous reviewers who is satisfied with the revisions done. I generally agree with the reviewer, but I still think that some extra revisions should be done before the article could be finally accepted for publication.

The main ones are:

the results for the GLMs are not adequately reported. Apart from giving the estimates and AIC values for the best model, the authors should run likelihood ratio tests (LRT) to give the statistics (i.e. chi square values, df, and p values) for each variable included in the best model and report them either in a table or in the text.Line 260-263- if you just omitted them from the analysis you are comparing data from fewer pantraps there that in other places, which could bias the result. You should standardize by subsampling (Gotelli and Colwell 2001) so you have the same number of pantraps to compare everywhere, as typically done. Please either do this subsampling or clearly justify why you do not do it. It does not seem correct to compare places with different sampling effort.

Others:

add letters for significant differences after post hoc analysis in fig 2. And complete the legend with this.Add reference for Shannon diversity index (line 240 )Add also a reference in the sentence given in lines 67-69.There is abundant literature regarding the effect of floral diversity on bee communities and it should be cited in the discussion section. Same for other parts, the discussion section lacks references to previous work. Please submit your revised manuscript by Aug 21 2022 11:59PM. If you will need more time than this to complete your revisions, please reply to this message or contact the journal office at plosone@plos.org. Please include the following items when submitting your revised manuscript:A rebuttal letter that responds to each point raised by the academic editor and reviewer(s). You should upload this letter as a separate file labeled 'Response to Reviewers'.A marked-up copy of your manuscript that highlights changes made to the original version. You should upload this as a separate file labeled 'Revised Manuscript with Track Changes'.An unmarked version of your revised paper without tracked changes. You should upload this as a separate file labeled 'Manuscript'.If applicable, we recommend that you deposit your laboratory protocols in protocols.io to enhance the reproducibility of your results. Protocols.io assigns your protocol its own identifier (DOI) so that it can be cited independently in the future. For instructions see: https://journals.plos.org/plosone/s/submission-guidelines#loc-laboratory-protocols. Additionally, PLOS ONE offers an option for publishing peer-reviewed Lab Protocol articles, which describe protocols hosted on protocols.io. Read more information on sharing protocols at https://plos.org/protocols?utm_medium=editorial-email&utm_source=authorletters&utm_campaign=protocols.

We look forward to receiving your revised manuscript.

Kind regards,

Amparo Lázaro, PhD

Academic Editor

PLOS ONE

Journal Requirements:

Reviewers' comments:

Reviewer's Responses to Questions

**Comments to the Author**

1. If the authors have adequately addressed your comments raised in a previous round of review and you feel that this manuscript is now acceptable for publication, you may indicate that here to bypass the “Comments to the Author” section, enter your conflict of interest statement in the “Confidential to Editor” section, and submit your "Accept" recommendation.

Reviewer #1: All comments have been addressed

2. Is the manuscript technically sound, and do the data support the conclusions?

Reviewer #1: Yes

3. Has the statistical analysis been performed appropriately and rigorously? 

Reviewer #1: Yes

4. Have the authors made all data underlying the findings in their manuscript fully available?

Reviewer #1: Yes

5. Is the manuscript presented in an intelligible fashion and written in standard English?

Reviewer #1: Yes

6. Review Comments to the Author

Reviewer #1: This paper presents an interesting study on the occurrence bees in urban and suburban habitats. This topic is still understudied, so the authors make an important contribution to recognizing these issues. I am pleased to accept the changes made to the manuscript and recommend publication of this manuscript.

7. PLOS authors have the option to publish the peer review history of their article (what does this mean?). If published, this will include your full peer review and any attached files.

Reviewer #1: No

---

## [Author Response · Author response to Decision Letter 1]

27 Jul 2022

To the PLOS ONE Editorial Board, 

Thank you for your thoughtful commentary. We found that the reviewer and editor comments helped strengthen our manuscript and we are happy to resubmit for continued review. We are attaching here a detailed comment-by-comment response to the reviewer and editorial suggestions. All issues have been addressed in text. 

One point that we would like to address is the author order. I, Israel Del Toro, as lead PI will remain the corresponding author for this work, however the student on the project Maggie Anderson should be first author. If possible I would like to be listed at the final author in the list of contributing members. 

Sincerely, 

Israel Del Toro

---

## [Editor Report · Decision Letter 2]

15 Aug 2022

PONE-D-21-32723R2B.Y.O. Bees: managing wild bee biodiversity in urban greenspacesPLOS ONE

Dear Dr. Del Toro,

Thank you for submitting your manuscript to PLOS ONE. After careful consideration, we feel that it has merit but does not fully meet PLOS ONE’s publication criteria as it currently stands. Therefore, we invite you to submit a revised version of the manuscript that addresses the points raised during the review process.

Although the authors have done a good effort to include my last comments, the statistical issue is still not resolved and therefore I cannot accept the manuscript for publication yet in its current form. In my previous comment I indicated that the results for the GLMs were not adequately reported. Apart from giving the estimates and AIC values for the best model (which one gets with the function ‘summary’), the authors should run likelihood ratio tests (LRT) to give the statistics (i.e. chi square values, df, and p values) for **each variable included in the best model **and report them either in a table or in the text. This can be obtained with function ‘Anova’ in library (car) or alternatively with function ‘Drop1’ applied to the best-fitting model. Otherwise we cannot know whether the variables that appear in the best model have a significant effect on the response or not. The authors have conducted two likelihood ratio tests to compare the best-fit GLMs with the global GLMs for both bee abundance and bee species richness, but this is not what I meant them to do.

Another important issue is that I see that in the last revised version the authors have removed a paragraph in the statistical analyses section where the GLMs were explained. Right now, there is no explanation of the GLMs conducted. The authors should explain clearly the statistical analyses conducted (type of model, error distribution, whether it is mixed model or not, predictor and response variables) and also how did they perform model selection (did they use automatic selection with dredge function in MuMIn package? If not, did they try all the possible combinations? How did they  do so?). Lastly, as commented above, the authors may add the LRT to the best-fitting model to show the significance of the variables included in it. Until all these issues are adequately solved I cannot accept the manuscript for publication.

We look forward to receiving your revised manuscript.

Kind regards,

Amparo Lázaro, PhD

Academic Editor

PLOS ONE

---

## [Author Response · Author response to Decision Letter 2]

30 Sep 2022

Comments attached in the cover letter.: 

Thank you for your last set of comments. This was a straightforward fix as we had inadvertently deleted a key paragraph that contained the missing details. This was easily resolved and updated. Thank you for catching this. We are hopeful that we can proceed to the next stage. 

Many thanks, 

Israel

---

## [Editor Report · Decision Letter 3]

10 Nov 2022

PONE-D-21-32723R3B.Y.O. Bees: managing wild bee biodiversity in urban greenspacesPLOS ONE

Dear Dr. Del Toro,

Thank you for submitting your manuscript to PLOS ONE. After careful consideration, we feel that it has merit but does not fully meet PLOS ONE’s publication criteria as it currently stands. Therefore, we invite you to submit a revised version of the manuscript that addresses the points raised during the review process. Although the authors have included the LRT as I indicated, there are a couple of minor issues that should be solved before publication: - in line 339-340, the authors indicate:  'the model’s goodness-of-fit was verified using likelihood ratio tests (Table 2)'. However, LRT do no test model fit, but the significance of the effect of different predictor variables on the response. So this sentence is incorrect. The authors might rather say that the significance of the predictive variables in the best-fitting model was tested using LRT.- Also in the new table 2 with the results of LRT, it might be 3 columns: df, Chi2 and p-values. It is fully correct to say in the legend of the table that these are the results of LRT, but in the column, the value of hte statistics is a value of Chi2 (not LRT). So please, change the heading of the column from LRT to Chi2. Also, p-values should be written adequately, using 3 decimals (or 4 if necessary), and using '< 0.0001' when p-values are smaller than 0.0001.

We look forward to receiving your revised manuscript.

Kind regards,

Amparo Lázaro, PhD

Academic Editor

PLOS ONE
---

## [Author Response · Author response to Decision Letter 3]

17 Jan 2023

1) in line 339-340, the authors indicate: 'the model’s goodness-of-fit was verified using likelihood ratio tests (Table 2)'. However, LRT do no test model fit, but the significance of the effect of different predictor variables on the response. So this sentence is incorrect. The authors might rather say that the significance of the predictive variables in the best-fitting model was tested using LRT.

Author response- Line 336-337 now reads: “the R package mass (Venables and Ripley, 2002) and the significance of the predictor variables in the best-fitting model was tested using likelihood ratio tests (Table 2).”

2) Also in the new table 2 with the results of LRT, it might be 3 columns: df, Chi2 and p-values. It is fully correct to say in the legend of the table that these are the results of LRT, but in the column, the value of hte statistics is a value of Chi2 (not LRT). So please, change the heading of the column from LRT to Chi2. Also, p-values should be written adequately, using 3 decimals (or 4 if necessary), and using '< 0.0001' when p-values are smaller than 0.0001.”

Author response- Table 2 column heading has been appropriately adjusted to Chi2, and p-value significant figures were extended as suggested by this reviewer.

---

## [Editor Report · Decision Letter 4]

25 Jan 2023

B.Y.O. Bees: managing wild bee biodiversity in urban greenspaces

PONE-D-21-32723R4

Dear Dr. Del Toro,

We’re pleased to inform you that your manuscript has been judged scientifically suitable for publication and will be formally accepted for publication once it meets all outstanding technical requirements.

Kind regards,

Amparo Lázaro, PhD

Academic Editor

PLOS ONE
---

## [Editor Report · Acceptance letter]

3 Apr 2023

PONE-D-21-32723R4 

B.Y.O. Bees: managing wild bee biodiversity in urban greenspaces 

Dear Dr. Del Toro:

I'm pleased to inform you that your manuscript has been deemed suitable for publication in PLOS ONE. Congratulations! Your manuscript is now with our production department. 

Kind regards, 

on behalf of

Dr. Amparo Lázaro 

Academic Editor

PLOS ONE